# The Relationship between Certain Parental/Household Socio-Economic Characteristics and Female Adolescent Obesity in Montenegro

**DOI:** 10.3390/children10050820

**Published:** 2023-04-30

**Authors:** Pavle Malovic, Erol Vrevic, Dragan Bacovic, Danilo Bojanic, Milovan Ljubojevic

**Affiliations:** Faculty for Sport and Physical Education, University of Montenegro, 81400 Niksic, Montenegro

**Keywords:** adolescents, nutrition assessment, obesity, socio-economic status, Montenegro

## Abstract

Background: Considering that obesity is characterized today as a public health challenge and an epidemic in many countries in the world and that one of the main predictors for obesity is socio-economic status (SES), the aim of this paper was to assess the relationship between the SES of parents/guardians and female adolescent obesity in Montenegro. Methods: A stratified random probability sample method was used, and the number of participants in this study was 596, aged 15.8 ± 0.58, from all three regions in Montenegro. As SES was a factor in this research, specific SES parameters such as household wealth and parental educational level were collected for parent/guardian of each child. The following anthropometric indices were utilized to evaluate nutritional status: body mass index (BMI) and waist to height ratio (WHtR). Results: Regarding nutritional status, it can be said that no statistically significant difference between female adolescents according to the regions of Montenegro was found. Of all the adolescents in the study, 15.4% of them were above the normal nutrition level as measured by BMI, while 12.2% were classified as obese by the WHtR. Furthermore, the study found a significant negative relationship between a mother’s level of education and obesity in female adolescents, with odds ratios of 0.31 (*p* = 0.035) and 0.19 (*p* = 0.009) for secondary and high level education, respectively. This suggests that daughters of mothers with higher levels of education are less likely to be obese. Conclusions: In regard to the nutritional status of the respondents in this study, their values fell within the normal range compared to the European average. However, the results regarding the relationship between certain SES characteristics and obesity suggest a similarity to developed countries.

## 1. Introduction

Today, obesity represents a public health challenge [1], and in many countries around the world, the issue is characterized as an epidemic [2]. Additionally, many authors consider obesity to be one of the factors that increases the risk for the occurrence of non-communicable diseases [3,4], as well as one of the basic predictors of a person’s health. It has been proven that adolescents with a high birth weight have a 1.93-times higher chance of facing overweight and obesity during their adolescence [5], and if obesity is present at an earlier age, there is a risk that the child may have problems with obesity during adulthood [6,7,8,9]. The important fact is that overweight and obese adolescents may be affected by psychosocial sequelae due to dissatisfaction with body shape, which includes depression, social stigmatization, bullying, behavioral problems, a lack of quality of life, self-esteem problems, etc. [10,11]. Additionally, mental health problems and factors may be associated with bad eating habits, which may lead to anorexia and bulimia nervosa, binge-eating disorders, or night eating syndrome [12,13,14,15]. The problems of bullying and stigmatization that are related to obesity in early childhood can also have an impact on physical and emotional health during adulthood [10]. Black et al. stated that the abovementioned sequelae are specifically significant for women because overweight and obesity can possibly be transmitted to the next generation (cited in [16]), which indicates that we should pay special attention to women as a population, due to their importance regarding reproductive functions. Therefore, if obesity is present in a woman during pregnancy, it can negatively affect the health of the both the mother and the child later in life [17]. Additionally, Rahman et al., Khan et al., and Machado emphasize the importance of tackling this problem in the female population because women suffering overweight and obesity are more likely to have complications related to pregnancy, such as gestational diabetes and hypertension, postpartum hemorrhage, preeclampsia, caesarian section, infection at the site of intervention, congenital malformations, birth before the estimated period, and the possibility of prenatal death of newborn babies (as cited in [16]). Regarding children and adolescents aged 5–19, the data show that the prevalence of overweight and obesity dramatically increases. Globally, over 340 million children and adolescents aged 5–19 were overweight or obese in 2016, while 18% of girls were overweight and 6% were obese, which is significantly more than in 1975. Additionally, in 2016, around 40% of women aged 18 and older were overweight, and 15% of these were obese, a number that has almost tripled since 1975 [18]. When it comes to Montenegro specifically, research conducted by Vasiljevic [19] demonstrated that 7.6% of female adolescents belong to categories over a normal weight as measured by body mass index, while 9.7% were considered obese in terms of waist to height ratio, which indicates that something needs to be done to address this problem. It is important to note that one of the key predictors for obesity is socio-economic status (SES) and economic insecurity [20], while financial status, as well as the level of parental education, are the most relevant indicators of SES, and may affect the risk of obesity in children [21]. Additionally, an important point by Hiilamo et al. [22] and Barich et al. [23] is that many studies have shown that the female population is more susceptible to the negative consequences of obesity caused by socio-economic status than the male population. Given that a positive ratio of overweight and obesity with SES has been confirmed in many developing countries, which the World Bank defines as countries with a per capita income of up to USD 12.275 [24], which also includes Montenegro [25], it can be said that a real possibility exists that Montenegro may be following the same trend. The high prevalence of obesity and overweight in developed, as well as developing, countries [2,3,4] has led many researchers to join the fight against this disease. An important method required to undertake this involves, in particular, a proper and adequate assessment of the condition of the target population. Specifically, a way of assessing the health of a particular demographic is the regular anthropometric control and monitoring of the conditions of respondents, which could be the key to preventing the rise of obesity as a common problem in human society. Therefore, on the basis of the aforementioned factors, the goal of this research was to determine the nutritional status of participants and to uncover the relationship between SES and female adolescent obesity in Montenegro, while the respondents are young and still able to make essential changes in their lives when it comes to nutritional status and nutritional habits if the results show that it is necessary.

## 2. Materials and Methods

The stratified random probability sample method was used according to the standards of national studies, and the total number of individuals examined in this study was 596 (aged 15.8 ± 0.58). The schools sampled for this research were determined using the Probability Proportional to Size (PPS) procedure and the Active Data option in Excel and included all three regions of Montenegro. Considering that in this research, household wealth and parental educational level are taken into account, the same number of questionnaires was collected. Regarding the socio-economic status of the parents/guardians of the respondents included in this study, it is defined on the basis of two questions, i.e., categories, which best determine the mentioned status [21] within the standardized questionnaire (European childhood obesity surveillance initiative—COSI) [26], and Table 1 shows the modified categorization of the replies offered.

Information concerning socio-economic status and levels of parental education was self-reported by the parents/guardians, and it is important to mention that the level of education was obtained for both parents/guardians where possible.

Measurements were carried out in accordance with the guidelines of the International Society for the Advancement of Kinanthropometry (ISAK). Experienced researchers, who are also professors and teaching associates at the Faculty for Sport and Physical Education at the University of Montenegro, were responsible for conducting the testing. The measurements were taken in the morning hours, so as not to encounter possible variations in certain parts of the body [27]. The measurements were recorded by an assistant who was solely in charge of maintaining accurate data entry [28]. The testing of the participants was conducted in the gymnasiums of secondary schools in Montenegro. Respondents were barefoot and wearing sports clothes during the measurement process. Body height was measured using an anthropometer, calibrated at 1mm. Body mass was measured using a digital scale with a precision of 0.1 kg. Waist circumference was measured using an inch strip, calibrated at 0.1mm, by precisely determining the measurement point, which was the area between the lower edge of the tenth rib and the iliac ridge, and was measured on bare skin. On the basis of these measurements, the following anthropometric indices were created to evaluate nutritional status: through body mass index (BMI percentile) and the waist to height ratio (WHtR). Reference categorization was used as the method recommended by the Center for Disease Control and Prevention (CDC). When assessing nutritional status, respondents who were below the 5th percentile were considered to fall into the underweight category; participants were placed in the normal weight category if values ranged from the 5th to the 85th percentile; while the overweight category included those respondents who were in the >85th to ≤95th percentile. At the very end of the scale, obesity was established if a value exceeded the 95th percentile [29]. Additionally, the reference category for the WHtR was 0.5, which means that all participants with values below 0.5 belonged to the non-obese category, while those with values of 0.5 and above belonged to the obese category [30].

The chi-square (χ^2^) test was used to determine possible differences in tested variables between the tested groups of respondents in the predefined nutrition categories, all in relation to regions. The level of significance was set to *p* < 0.05. Binary logistic regression analysis was used to access the relationship between socio-economic status and obesity indexes. Connectivity is presented as an odds ratio (OR) with a confidence interval of 95% (CI—confidence interval) and a statistical significance of *p* < 0.05. In binary logistic regression models, the dependent variable was the WHtR ratio and was encoded as a dichotomy variable with reference values set at 0.5.

## 3. Results

This study involved 596 female adolescents from secondary schools in Montenegro, with an age range of 14.4 to 17.5 years and a mean age of 15.8 ± 0.58 years. All participants completed the assessments, and Table 2 presents the descriptive parameters of their body height, body mass, and waist circumference. Anthropometric indices were calculated based on these measurements to evaluate the nutritional status.

Regarding the socio-economic characteristics of parents/guardians, Table 3 presents the precise information regarding the data collected. Results show that the majority of mothers and fathers had a secondary level of education (65.6% and 65.4%, respectively), while 88.6% of respondents reported a higher level of household economic status.

Based on the results shown in Table 4, it should be noted that female adolescents, according to the BMI percentiles represented as percentage values, mostly belong the normal weight category. Most of the respondents that crossed the line into above normal weight are from the territory of the central region of Montenegro. When examining the general level of nutrition of female adolescents, it should be noted that 1.7% belong in the underweight category, 82.9% belong in the normal weight category, 11.9% belong in the overweight category, and 3.5% belong in the obese category. In total, 15.4% belong to categories above normal weight. When analyzing the data obtained through χ2, it should be noted that there were no statistically significant differences between the respondents from different regions in terms of BMI-percentiles. When considering the results regarding WHtR ratio, it can be seen that the female respondents, in the overall sample, mostly belong to the normal weight category, i.e., their values are below the overweight limit (0.50), accounting for 523 respondents (87.8%), while 73 participants belong to the obese category (12.2%). When regions are taken into account, it can be said that the highest number of respondents who are suffering from obesity were in the central region, which means that out of 275 respondents from this area, 41 were considered obese (14.9%); out of 175 respondents from the northern region, 18 were considered obese (10.3%); and in the coastal region, out of 146 respondents, 14 were considered obese (9.6%). Regarding all three regions, most of the respondents from the coastal region belonged to the normal weight category: 90.4%. The data obtained through χ2 analysis indicates that there were no statistically significant differences in WHtR ratio among the tested respondents from different regions.

Based on the results shown in Table 5, a statistically significant relationship between the mothers’ education levels and female adolescent obesity was observed. Specifically, respondents whose mothers had a mid-level of education were 69% less likely to be obese (OR = 0.31; *p* = 0.035), while respondents whose mothers had higher levels of education were 81% less likely to be obese (OR = 0.19; *p* = 0.009), relative to the reference category.

## 4. Discussion

Regarding nutritional status, as can be seen in the first two tables, it can be said that there is no statistically significant difference between female adolescents in terms of the regions of Montenegro. However, regarding BMI, 11.9% of respondents were found to be overweight, while of these, 3.5% were obese. The total number of adolescents above the normal nutrition level was 15.4%. When we take into account data from a PhD dissertation by Vasiljevic [19] on a sample of adolescents in Montenegro, it is clear that the percentage of adolescents suffering from overweight and obesity is much higher in the current study—in their research, 6% of respondents were overweight, while only 1.6% were obese. If we make a comparison with the United States, there has been a dizzying rise in obesity from decade to decade. From 1980 to 2014, obesity prevalence had increased from 10% to 21% [31], while other official data have confirmed that between 2017 and 2018, 16.1% of adolescents were overweight, 19.3% were obese, and 6.1% belonged to the extreme obesity category [32]. In Asian countries, the prevalence of overweight among female adolescents was 13.7%, while obesity was confirmed to be 6,2% [33]. Regarding African countries, it should be mentioned that over the years, overweight and obesity have increased. From 1998 to 2016, the trend toward overweight and obesity in female adolescents increased from 26.77% to 33.88%, respectively [34]. Additionally, a recent publication, which, among other things, calculated obesity prevalence assessment based on the value of BMI, provides an accurate picture of the level of nutrition in a sample of adolescents aged 15 and over in 35 European countries, including the 27 European Union (EU) Member States, 5 EU candidate countries, and 3 countries located in the European Free Trade Association (EFTA). In this research, it is confirmed that the mean value of obesity in all conducted countries for female adolescents was 15%. Thus, it can be said that the girls in this study (15.4%) were above the defined value at the European level. It would be interesting to compare the results obtained in this study with results from countries that surround Montenegro. Thus, considering that the corresponding levels are 15% in Serbia, 13% in Croatia, 16% in Slovenia, and 17% in North Macedonia [35], it can be said that the results are fairly similar. It is important to mention that the aforementioned countries were all part of the former Yugoslavia, and it can therefore be assumed that they likely have similar lifestyles. However, Montenegro is not included in this list, so the data obtained in this research gain further significance.

Results based on the WHtR ratio indicate that 12.2% of female adolescents belonged to the obese category. In research conducted by Vasiljevic [19], based on the aforementioned index, 9.7% of female respondents were obese. However, it should be taken into account that the sample in that study included 771 female adolescents of all secondary school levels, while the current study included 596 respondents from only the first and second levels of secondary school, so there is a real possibility that more respondents from specific grades could reflect more realistic conditions.

Based on the above, it can clearly be seen that there is a difference between the resulting values when comparing these two indicators. However, a main limitation of BMI is that it does not separate muscle tissue from fat tissue in the calculation nor the distribution of fat across certain body parts [36]. Therefore, many authors argue that the WHtR ratio should be used before BMI, for many reasons including the aforementioned ones [37,38,39,40,41,42,43,44].

Based on the results obtained through the assessment of the relationship between the SES of parents/guardians and adolescent obesity, it can be seen that the situation is not expected and deviates from previously stated claims. Specifically, one of the most important systematic review studies [45] found that there was a negative ratio of obesity and SES in developed countries in people of the female sex, i.e., an increase in SES decreases the prevalence of obesity. Additionally, recent studies conducted in the United States and Spain have confirmed the negative ratio of SES and obesity in adolescents, but in this case, it should also be noted that the US and Spain are developed countries [46,47]. In contrast, studies conducted in Africa confirmed that the relationship between socio-economic status and adolescent obesity is positive and that adolescents from the highest SES households had a greater chance of being obese then those from lower SES households. They posited that an increase in wealth increases the availability of unhealthy food [34,48]. However, this study found that the level of a mother’s education has a significantly negative relationship with obesity of female adolescents, and it can be said that the situation is the same as in other developed countries. It has also been reported that the results of the relationship between the level of household wealth and the obesity of female adolescents are following a similar trend. Still, in this case, a statistically significant relationship was not found. Therefore, the fact is that in most households, mothers have the responsibility when it comes to food choices and culinary-based obligations [49,50]. Based on the above, there is a very real possibility mothers having a higher level of education as well as the increasing availability of information via social media, when used constructively, can influence the formation of a clearer picture of the importance of healthy lifestyles that should be practiced in children in which the inevitable factor is the consumption of healthy food in optimal quantities. The study design represents a limitation of this investigation, as it does not take into account more covariates that determine nutritional status, physical activity, and the diet and daily habits of adolescents. Additionally, more characteristics of parental socio-economic status could be taken into account. Moreover, during the realization of this study, researchers faced interference from the COVID-19 pandemic, e.g., a limitation for respondent sampling. All these limitations need to be taken into account for future research on the same topic. However, these limitations do not diminish the importance of this preliminary study in Montenegro. Instead, it serves as an excellent starting point for future research, which can be able to access more detailed data and reach more thorough conclusions.

## 5. Conclusions

Based on the above, it is possible to define the final conclusions of this study with a high level of confidence. Specifically, the study’s findings have practical implications in terms of raising awareness about the relationship between SES and obesity in female adolescents. It is recommended that the results be presented to parents/guardians through public meetings or stands, as they play a significant role in their children’s nutritional status [51]. Considering Montenegro has been working towards European and Euro-Atlantic integration for a long time, where one of the benefits of would be the better socio-economic status of citizens [52], may be a reason for the negative correlation between SES and obesity. It can be assumed that the country is on track to meet the requirements and preconditions for joining the society of developed countries. While keeping in mind the limitations and recommendations provided in this study, these findings may serve as a baseline for further research on the relationship between SES and adolescent overweight and obesity in southeastern European countries.

## Figures and Tables

**Table 1 children-10-00820-t001:** Questions and predefined answer options are included in the standardized COSI questionnaire for collecting data on the socio-economic status of parents/guardians as well as categorization of response options used for the purposes of this research.

Questions	Answers Provided in the Questionnaire	Categorization of Replies for Research Purposes (Merged Responses)
Parental education level		
What is the highest level of education you or your spouse/partner have completed? Please select one answer only for each of you.	Primary school or less Secondary/High school Vocational school Undergraduate/Bachelor’s degree Master’s degree or higher	Basic level of education Secondary level of education (2nd and 3rd responses) Higher level of education (4th and 5th responses)
Economic status level		
Including all your household earnings per month, please tick the box which best represents your household situation? Please tick one box.	We easily pass the month with our earnings We pass the month without serious problems with our earnings We have trouble meeting the ends the month with our earnings We barely meet the ends in the month with our earnings	Lower level of household economic status (3rd and 4th responses) Higher level of household economic status (1st and 2nd responses)

**Table 2 children-10-00820-t002:** The anthropometric characteristics of the participants (N = 596).

Measures	Min	Max	Mean ± S.D.
Age (years)	14.4	17.5	15.8 ± 0.58
Body height (cm)	148.0	189.9	167.67 ± 6.34
Body mass (kg)	42.4	134.6	61.19 ± 10.74
Waist circumference (cm)	58.9	123.0	73.41 ± 8.68

Legend: Min—minimum; Max—maximum; Mean ± S.D.—mean ± standard deviation.

**Table 3 children-10-00820-t003:** Level of parental education and household economic status.

Scheme	Collected	
	N	%
Mother’s education level		
Basic level of education	21	3.6
Secondary level of education	387	65.6
Higher level of education	182	30.8
Father’s education level		
Basic level of education	17	2.9
Secondary level of education	377	65.4
Higher level of education	183	31.7
Level of household economic status		
Lower level of household economic status	67	11.4
Higher level of household economic status	523	88.6

**Table 4 children-10-00820-t004:** Obesity prevalence in terms of regions based on BMI percentile and WHtR ratio in female adolescents and possible differences between them (N = 596).

Region	BMI Percentile										*p*
	Underweight		Normal Weight		Overweight		Obesity		Total		
	N	%	N	%	N	%	N	%	N	%	
Northern	3	1.7	148	84.6	18	10.3	6	3.4	175	100	0.743
Central	6	2.2	223	81.1	34	12.3	12	4.4	275	100	
Coastal	1	0.7	123	84.2	19	13	3	2.1	146	100	
Total	10	1.7	494	82.9	71	11.9	21	3.5	596	100	
Region	WHtR										*p*
	Non-obese				Obese		Total				
	N		%		N	%	N		%		
Northern	157		89.7		18	10.3	175		100		0.183
Central	234		85.1		41	14.9	275		100		
Coastal	132		90.4		14	9.6	146		100		
Total	523		87.8		73	12.2	596		100		

Legend: BMI percentile—percentile values of body mass index; WHtR—waist to height ratio; Region—region in Montenegro; *p*—statistical significance.

**Table 5 children-10-00820-t005:** Relationship between socio-economic characteristics and obesity rates in female adolescents.

Socio-Economic Characteristics	OR (95%CI)	*p*
Mother’s education level		
Basic level of education	1	
Secondary level of education	0.31 (0.10–0.92)	0.035
Higher level of education	0.19 (0.06–0.66)	0.009
Father’s education level		
Basic level of education	1	
Secondary level of education	2.26 (0.43–11.84)	0.333
Higher level of education	2.11 (0.36–12.19)	0.405
Level of household economic status		
Lower level of household economic status	1	
Higher level of household economic status	0.81 (0.37–1.74)	0.586

Legend: OR (95%CI)—odds ratio (OR) with a confidence interval of 95%.

## Data Availability

All data are available in the private archive of the corresponding author.

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
