# Peer review of "The Relationship between Certain Parental/Household Socio-Economic Characteristics and Female Adolescent Obesity in Montenegro"

_children, 2023, doi:10.3390/children10050820_

Round 1

Reviewer 1 Report (New Reviewer)

I thought the scope of this paper and the analyses were insufficient for publication. You should consider more variables in your analysis rather than just weight status (which you termed 'nutritional status'). For example, you should consider physical activity and screen time or else some element of dietary intake. It was interesting to associate BMI with SES but you should have considered more participant characteristics such as maternal and paternal weight status. I noticed that in your statistical model, you did not consider any covariates, which is a huge flaw. In cross-sectional studies such as this you cannot imply causation when you find a significant relationship, just correlation, which is a limitation that you didn't mention.

If you considered more variables both your introduction and discussion would be more interesting. You did not list the limitations of your work in the discussion, I found your conclusion far too long and your bibliography was very outdated. I would expect most citations to be 0 to 5 years old with a few of them being up to 10 years old. In your paper most citations were greater than 10 years old.  You tried to compare your results with those from other studies, but you did not cite enough other studies to make the discussion interesting. 

To summarize, I would suggest redoing the study to include more variables of interest and to do a more thorough analysis with consideration of covariates. I also suggest that you compare with other studies not just 1 study in the US. Finally, I would recommend you get a good English editor to work on the translation 

Author Response

Reviewer 2 Report (New Reviewer)

I suggest carrying out a review according to the notes made in the article

Author Response

Dear reviewer, thank you for the comments and opinion about our research. Regarding your suggestion about English requirements, we used paid English editing service that MDPI offers, to make this paper more readable and grammarly perfect.

Reviewer 3 Report (New Reviewer)

Please fix grammatical errors and formatting through the text.

In title: please add "parental/household", before socio-economic.

In abstract:

Line 14: Please mention the mean±SD or range of age.

Please mention the sampling method.

Line 15 , 16 the sentence: “Regarding that SES of parents/guardians was considered, the same number of questioners was collected” is unclear. What kinds of data were gathered regarding the measurement of SES and which instruments were used for data gathering? There is not clear which criteria were used for evaluation of nutritional status.

Results: Results should be rewritten. There are no results regarding the SES situation and nutritional status.

Conclusion is not based on the presented results in the abstracts.

Introduction:

Previous studies showed that the moderate range of overweight among the adolescents in your country. So, it seems that the prevalence of obesity should be lower than overweight. How did you justify your proposed research questions?

In lines 47 to 54 you mentioned the side effects of female adolescent obesity and focused on childbearing and pregnancy complication. How about the behavioural and psych-social complications of overweight and obesity for adolescents? What about the eating behaviours?

Lines 60 to 62: There is no relation between these findings with your proposed research question.

Methods:

There are gaps on:

-        Study design

-        Setting of the study

-        Sample of the study

-        Sample selection methodology

-        Range of the age of the children

-        Among different socio-economic status you have just collected data regarding parents’ education and situation of household earning. In my opinion, in one hand, these are not enough for SES categorization and on the other hand, the variable you considered for SES calculation should be explained through the text.

Table 1 should be moved to result section.

Table 3 should be moved to result section. The right column could be deleted.

Line 144: Reference should be added.

Why you did not use BMI for age z-score for obesity and overweight categorization?

You can show the results of table 4, 5, and 6 in one table. You can add some general characteristics of the studied samples in one table. Or you can show the situation of obesity and SES and region!!!?? in one table by cross-tab analysis.

Discussion: needs to be rewritten.

Conclusion:

Line 278: How can you explain that these two variables that you mentioned are among the most important cause of adolescent’s obesity? Researches showed that there are different variables and determinants for adolescents’ obesity. Including food environment, peer influence, eating behaviours, and also some household characteristics.

Line 293: This conclusion is not based on the findings of this study.

The conclusion needs to be rewritten.

Author Response

Reviewer 4 Report (New Reviewer)

The manuscript entitled “ The impact of Some Socio-Economic characteristicshousehold 3 wealth and parental educational level on female adolescent 4 nutrition obesity in Montenegroo: a national study” I found the manuscript is well written and sound , some minor points that needs to be addressed for better presentation

-        In the abstract “ SES” : should mention the abbreviation beside the full name in the first sentence of the abstract , also BMI and WHtR were not defined

-        Line 35 : [9, cited in 10], ????

-        Considering the Impact of SES on obesity the level of a mother's education is significantly negatively linked to the obesity of female adolescents.

-        Line 36- 37: is not well presented and need rephrasing

-        Line 40 : how could you considered adolescents aged 5-19: is this right ? mention reference

-        What are the main causes of the dramatical increase in obesity in line 41? As you mentioned with suitable citation , I think this part needs revision to mention the full idea

-        Line 47 : what you mean with Waist to Height ratio? Mention in this regard

-        Table 6 : needs to address the illustration of the presented data in this table in the footnote

-         

-        The conclusion needs revision and rephrasing and you should add the major limitations of the study.

Round 2

Reviewer 1 Report (New Reviewer)

Overall, it is much improved. However, there are still English errors and parts of it were difficult to read.

Comments

Abstract:  use predictors instead of benchmarks. Rewrite lines 22-25 to make it clearer

Conclusion: Shorten to make it more concise. Delete the sentence from line 360-365. Rephrase the last sentence as your study won't necessarily make another one more 'easy' to conduct as you didn't develop a new method.

References: Remove the references from 1946 and 1956 - they are far too old

Author Response

Reviewer 3 Report (New Reviewer)

The authors tried to answer all comments. But still there are some points should be covered:

Extensive editing of English language and style required.

In abstract: You should just mention what kind of data were collected for measurement of SES?

Please rewrite this sentence: The number of adolescents above the normal nutrition level as measured by BMI...

Please rewrite the conclusion of the abstract.

Results should be start by some explanations on studied samples. In my opinion some tables could be merged.

In my opinion the design of the study is the most limitation for interpreting the results which authors should mention it.

Author Response

This manuscript is a resubmission of an earlier submission. The following is a list of the peer review reports and author responses from that submission.

Round 1

Reviewer 1 Report

Childhood obesity is a topic of interest for most countries. In this paper, authors present new data that highlight female adolescent obesity markers and the relationship with family SES. Although, examining data for Montenegro offers some insight into the correlates of obesity in this specific country the paper offers nothing new to the broader literature.  

Socioeconomic status and obesity has been widely studied across settings including in developed and developing nations.  It has been studied in longitudinal settings as well as with multiple measures.  It has been examined by gender, race and ethnicity.  In this study, the authors offer a cross-sectional, correlational study of gender, ses, and obesogenic indicators such as BMI and waist-to-height ratio with no other controls.   This examination did not offer any new insights into adolescent obesity.

I readily admit that I would struggle to write a paper in Montenegrin.  Thus, it is understandable that these authors took on a daunting task to write a scientific paper in English. However, the authors were challenged with the English language to the point that it severely impeded the readability of the paper.  One example is the interchanging of words that do not mean the same thing such as nutrition or nutritional status for obesity.  Grammar was a problem in every paragraph. The tables lacked specificity and clear formatting.

Overall, the combination of writing clarity issues and the lack of new insight into correlates of female adolescent obesity make this a very weak paper.

Reviewer 2 Report

Obesity is a disease that has been increasing in recent decades at all ages and is the precedent for the development of chronic diseases that are the cause of the main causes of mortality in the world.

I send the following comments to the authors for their consideration:

         1.       Who collected the information?

2.       It is mentioned that adolescents from the 2019-2020 school year were included in the study. How did the COVID 19 pandemic interfere in the data collection?

3.       Was the study approved by an Ethics and Research Committee?

4.      Was parental and adolescent consent and assent obtained?

5.       Was a sample size calculated? It is mentioned that the data was obtained from a national database and from three regions of Montenegro, why were only 596 adolescents included?

6.       How was the information collected from the questionnaire with the parents? Was it applied by a surveyor, or self-applied, was it online, etc.? It is necessary to place this information in methodology.

7.       Why include only female adolescents?

8.       Place in the methodology that the BMI percentiles were obtained, since it only mentions that the crude BMI was obtained, likewise, it is necessary to place the referent, was the CDC or WHO used? The categories of how the adolescents were classified as normal weight, overweight or obese should be added.¿Cómo categorizaron la condición nutricia?

9.       It is suggested to add an initial table that identifies the baseline characteristics of the study population, for example mean age, level of parental education, level of education, mean weight, height, waist of all participants.

10.   It is suggested that researchers add in methodology how they classified the WHtR variable, that is, the cut-off points.

11.   It is necessary to include in the methodology that the level of education was obtained for both parents.

12.   It is suggested that before generating table 4 where the logistic regression model is, a bivariate table be built between the categories of obesity and socioeconomic level and level of education, this in order to evaluate if it is appropriate to build a model of regression.

13.   In table 4 add the confidence intervals of the OR.

14.   In the description of results, the researchers describe that “When it comes to the level of father's education, a double chance to be obese was for respondents whose fathers had a secondary (OR=2.26) and a higher level of education (OR= 2.11), compared to the reference category.” However, no data is significant in the model. The same is true for the economic level of the household.

15.   In Table 4, why did you use WHtR as the dependent variable? Why was the categorical variable of BMI percentiles not used?

16.   Was the regression model adjusted for any variable?

17.   What are the strengths and limitations of the study? In addition to the fact that the study was carried out only in women.

Round 2

Reviewer 1 Report

Thank you to the authors for responding to my first comments and making improvements to the manuscript. I continue to assert that this paper offers nothing new to the literature.  If the argument is that Montenegro offers a glimpse into developed vs non-developed country results concerning adolescent female obesity and SES, then that argument needs to be strengthened in the front part of the paper.  It is an uphill battle as Montenegro is highly rated on the Human Development Index.  I still do not see the clear linkages to economic status in the results section -- why is this not highlighted and presented first if the focus of the paper is on SES and obesity?  Similiarly, I do not see the reasoning behind the comparisons between regions.  Are regions proxies for economic development?  Writing continues to be a problem.  One cannot rely on a journal to fix writing issues as the authors indicated they would do for this paper.  Without a clear thesis that tests something new about obesity and SES, this paper does not contribute to the body of literature. 

Reviewer 2 Report

The authors discussed and responded appropriately to each of the comments.